# HypBrain: Hyperbolic Space Guided Cross-Subject Vision-Brain Representation Learning Framework

## Abstract

Understanding the intricate mappings between visual stimuli and their corresponding neural responses is a fundamental challenge in cognitive neuroscience and artificial intelligence. Current vision-brain representation learning approaches predominantly align paired images and functional magnetic resonance imaging (fMRI) responses within a shared Euclidean embedding space. However, Euclidean geometry struggles with the exponential complexity of visual/neural hierarchies, resulting in semantically undiscriminating embeddings. To overcome this, we propose HypBrain, a novel framework that leverages hyperbolic geometry to learn a shared, cross-subject vision-brain representation. Our framework maps both visual information and multi-subject fMRI responses into a shared Lorentz model, a geometry uniquely suited for embedding hierarchical data. We introduce a new mapping logic where abstract visual concepts are embedded near the hyperbolic origin, while more specific fMRI responses are situated in the exponentially expanding periphery, naturally capturing the "entailment" relationship between visual and neural data. Notably, we train a hyperbolic encoder on multi-subject fMRI data to integrate both common and unique characteristics of individual brain responses. Experimental results demonstrate that HypBrain not only exhibits robust capabilities in accurately quantifying semantic alignment but also achieves significant advancements in capturing cross-modal semantic relationships solely by optimizing the geometric properties of the embedding space. Our method confirms the superiority of hyperbolic geometry in aligning cross-modal semantic representations and modeling hierarchical associations, thereby offering an innovative perspective in the field of vision-brain representation learning.

## 1 Introduction

The brain, as the core of human cognition and perception of the world, constantly encodes the various external stimuli that we encounter daily. Recent advancements have significantly deciphered semantic information from brain responses to visual stimuli (Scotti et al., 2023; Takagi & Nishimoto, 2023a; Liu et al., 2024; Zhou et al., 2024; Wang et al., 2024a). These methods utilize functional magnetic resonance imaging (fMRI) (Logothetis, 2008)-acquired neural patterns to learn meaningful brain features, aligning them with image features from pretrained vision-language models (VLMs) to unravel how the brain interprets the visual world. However, existing models primarily employ a holistic, cross-modal learning strategy, often neglecting the intricate hierarchical semantic relationships between visual inputs and brain activity, thereby restricting effective structural modeling. In fact, image data inherently contains multi-scale semantics, ranging from low-level features like pixels and edges to high-level concepts such as complete objects and scenes. Likewise, fMRI signals exhibit a corresponding multi-layered structure, reflecting the brain's bottom-up, hierarchical processing from local regions handling basic features to whole-brain integration of advanced information (Miliotou et al., 2023; Chen et al., 2025a). Therefore, modeling image features and neural responses as hierarchical structures within an embedding space is essential for a more accurate reflection of the brain's visual encoding mechanisms.

However, Euclidean space struggles with hierarchical data due to its polynomial volume growth, which inadequately accommodates exponentially expanding hierarchical structures. This leads

to compressed low-dimensional concept embeddings and reduced semantic discriminability (Matoušek, 1999). As illustrated in Figure 1(a), while Euclidean distances show semantic similarity between concepts such as "animal" and "cat", they fail to capture the deeper hierarchical "is-a" relationship. Conversely, with its negative curvature, hyperbolic geometry offers a natural solution as its exponentially expanding representation space is ideal for embedding complex hierarchical information (Gromov, 1987; Sala et al., 2018). As shown in Figure 1(b), hyperbolic embedding effectively represents semantic hierarchies, positioning abstract concepts near the center and concrete ones in the expanding periphery. This property can model the vision-brain relationship. VLM-extracted image semantics are often more abstract than pixel data, while fMRI responses, as brain's physiological activity to visual stimuli, exhibit higher information specificity. Consequently, the fMRI embedding $F_{cat}$ for viewing a "cat" image is mapped to a position distant from the origin ("animal"), whereas the image embedding $I_{cat}$ is situated closer to the origin. Similarly, for an image containing "cat and dog", $F_{cat\ and\ dog}$ and $I_{cat\ and\ dog}$ are located further from the origin in hyperbolic space than embeddings for "cat" alone. Furthermore, consistent with hyperbolic geometry, fMRI patterns should be intrinsically encompassed within the semantic information of the image. In the projected view of Figure 1(b), $F_{cat\ and\ dog}$ lies at the intersection of the "cone" regions of $I_{cat}$ (i.e., the yellow region) and $I_{dog}$ (i.e., the purple region), indicating that the specific neural response to an image containing "cat and dog" is hierarchically entailed by the abstract concepts of both "cat" and "dog".

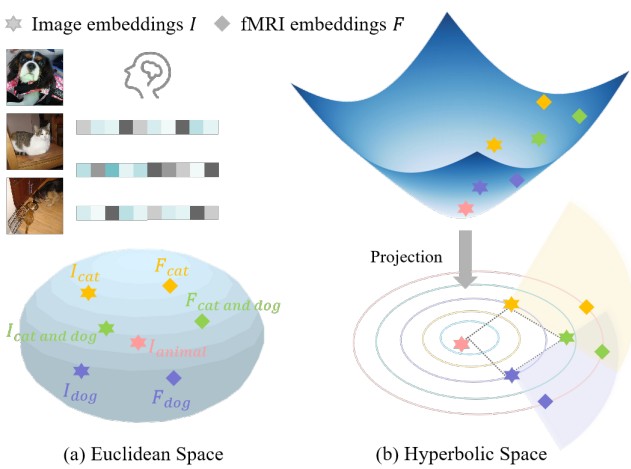

(a) Euclidean Space    (b) Hyperbolic Space

Figure 1: Conceptual comparisons of Euclidean embeddings and hyperbolic embeddings.

Hyperbolic geometry is particularly effective for learning hierarchical representations in diverse modalities such as images and text (Desai et al., 2023; Srivastava & Wu; Yang et al., 2024; Wang et al., 2024b; Pal et al., 2024; Gonzalez-Jimenez et al., 2025). However, its potential in modeling the rich semantic relationships between images and brain activity remains largely unexplored. To address this, we propose HypBrain, a hyperbolic space-based vision-brain representation learning method. HypBrain leverages hyperbolic geometry to align neural activity with image representations, explicitly modeling their inherent hierarchical structures. It represents concepts from both modalities using the Lorentz model (Nickel & Kiela, 2018). The learned embeddings are optimized via a novel hyperbolic contrastive loss combined with an entailment loss, capturing both semantic relationships and hierarchical dependencies. Crucially, our approach employs fMRI data from multiple subjects to train a dedicated hyperbolic encoder, enabling the model to learn shared neural response patterns across subjects while retaining subject-specific information, thus addressing a key challenge in cross-subject vision-brain learning.

To validate the effectiveness of the proposed method, we conduct experiments on the Natural Scenes Dataset (NSD) (Allen et al., 2022). Results indicate that HypBrain achieves remarkable success in accurately quantifying semantic alignment and capturing cross-modal semantic relationships by optimizing the geometric properties of the embedding space, outperforming state-of-the-art (SOTA) methods. Compared to their Euclidean counterparts, hyperbolic embeddings better capture semantic correlations and hierarchical relationships between different modalities, demonstrating significant

improvements in downstream tasks. Additionally, our framework performs exceptionally well across various VLMs, showcasing its strong versatility. Our contributions summarize as follows:

- We design a novel architecture with a hyperbolic tokenizer and shared encoder for multi-subject fMRI data, enabling simultaneous learning of individual and cross-subject patterns for generalized hyperbolic features.
- HypBrain leverages a hyperbolic fMRI encoder in conjunction with a frozen VLM to project neural responses and image features into a shared Lorentz manifold, precisely capturing intricate high-level semantic associations between visual stimuli and brain activity.
- To achieve effective cross-modal semantic alignment, we focus on semantic entailment relationships between neural representations and visual stimuli, enhancing the hierarchical discriminability of learned features.
- HypBrain consistently demonstrates either superior or comparable performance across various downstream tasks, which underscores the significant advantages of hyperbolic geometry in effectively modeling complex semantic hierarchies.

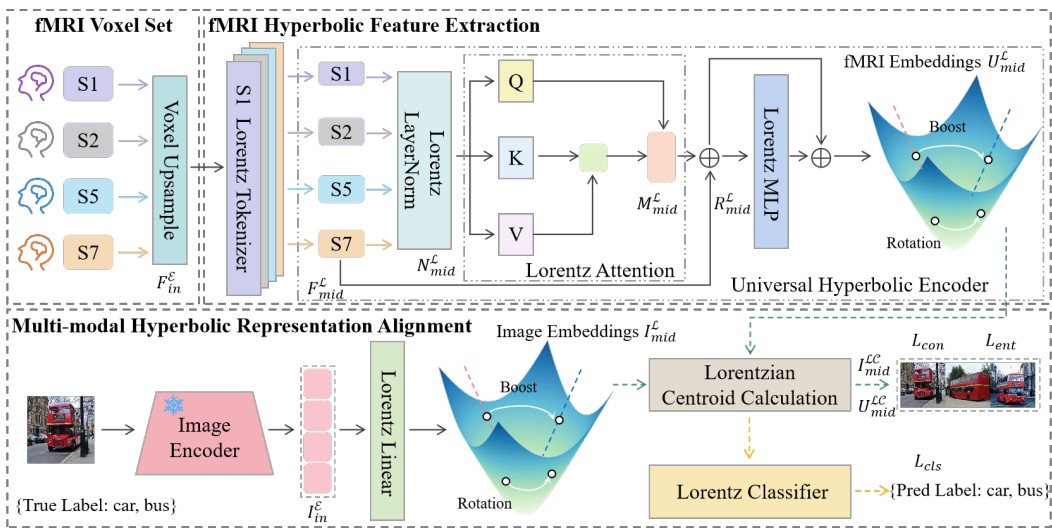

Figure 2: An overview of HypBrain. HypBrain is a cross-subject vision-brain representation learning framework. The framework consists of two main components: **fMRI Hyperbolic Feature Extraction** and **Multi-modal Hyperbolic Representation Alignment**. The former encodes different fMRI responses into unified hyperbolic embeddings through a cross-subject tokenizer and a universal hyperbolic encoder. The latter transforms image features extracted by a VLM image encoder into the Lorentz manifold, then semantically aligns them with the hyperbolic fMRI embeddings.

## 2 METHODOLOGY

We propose a novel cross-subject vision-brain representation learning scheme, HypBrain, which explicitly captures complex semantic relationships by leveraging the intrinsic hierarchical structure of both fMRI and image data. As shown in Figure 2, our method aligns latent semantic representations from different modalities within a unified hyperbolic space and comprises two key components: fMRI hyperbolic feature extraction and multi-modal hyperbolic representation alignment. For details on hyperbolic geometry and its Lorentz model, which forms the foundational framework for this study, please refer to Appendix B.

### 2.1 FMRI HYPERBOLIC FEATURE EXTRACTION

To address inter-subject brain activity variability (Gordon et al., 2017) and enhance model generalization, we introduce a novel fMRI encoder. This encoder learns hyperbolic embeddings from multi-subject data. Given the diverse fMRI voxel sizes (Finn et al., 2017), we standardize dimen-

sions across datasets using interpolation-based upsampling. The architecture consists of a cross-subject hyperbolic tokenizer, which extracts subject-specific neural representations, and a universal hyperbolic encoder, which captures shared response patterns.

### 2.1.1 CROSS-SUBJECT HYPERBOLIC TOKENIZER

Initial brain signals $F_{in}^{\mathcal{E}} \in \mathbb{R}^{1 \times (v+1)}$, with $v$ as the unified voxel dimension, are projected into a hyperbolic manifold using a Lorentz linear layer. This generates a rich token representation, denoted as $F_{in}^{\mathcal{L}} \in \mathbb{L}^t, \mathbb{R}^{(t+1) \times (v+1)}$. To capture subject-specific information, $F_{in}^{\mathcal{L}}$ is then fed into a subject-specific tokenizer, yielding the target embedding $F_{mid}^{\mathcal{L}} \in \mathbb{L}^d, \mathbb{R}^{t \times (d+1)}$. Specifically, the spatial component $F_{in-space}^{\mathcal{L}}$ is extracted from $F_{in}^{\mathcal{L}}$, linearly transformed, and concatenated with the recomputed temporal component $F_{in-time}^{\mathcal{L}}$ to form the subject-specific fMRI embedding $F_{mid}^{\mathcal{L}}$. Our operation, diverging from a full Lorentzian transformation (Chen et al., 2021), is defined as follows:

$$F_{\text{in-time}}^{\mathcal{L}} = \sqrt{\frac{1}{c_{\text{in}}} + \|W^T F_{\text{in-space}}^{\mathcal{L}} + b\|_2^2} \tag{1}$$

$$F_{\text{mid}}^{\mathcal{L}} = [F_{\text{in-time}}^{\mathcal{L}}, W^T F_{\text{in-space}}^{\mathcal{L}} + b] \cdot \sqrt{\frac{c_{\text{mid}}}{c_{\text{in}}}} \tag{2}$$

Here, $W$ represents the weight matrix, $b$ is the bias term, and $c_{in}$ and $c_{mid}$ denote the curvature parameters of the input and output manifolds, respectively.

### 2.1.2 UNIVERSAL HYPERBOLIC ENCODER

The universal hyperbolic encoder maps brain semantic embeddings across subjects into a shared latent space. We employ a Lorentz geometry-based Transformer encoder for fMRI representations.

**Lorentz Layer Normalization.** To maintain the properties of fMRI embedding $F_{mid}^{\mathcal{L}}$ in the hyperbolic manifold, we apply a Lorentz normalization layer, as described in (Yang et al., 2024). The spatial component $F_{mid-space}^{\mathcal{L}}$ undergoes standard layer normalization and is then concatenated with the temporal component $F_{mid-time}^{\mathcal{L}}$ to obtain the new hyperbolic feature, $N_{mid}^{\mathcal{L}}$.

**Lorentz Multi-head Self-attention.** Within the encoder, the Lorentz multi-head self-attention mechanism (Chen et al., 2021) is crucial for capturing intricate fMRI feature dependencies in hyperbolic geometry. $N_{mid}^{\mathcal{L}}$ is linearly transformed into query $(Q)$, key $(K)$, and value $(V)$ spaces, subsequently projected as points onto the Lorentz manifold, expressed as:

$$Q^{\mathcal{L}} = N_{\text{mid}}^{\mathcal{L}} \otimes^{c_{\text{mid}}} W^Q, \quad K^{\mathcal{L}} = N_{\text{mid}}^{\mathcal{L}} \otimes^{c_{\text{mid}}} W^K, \quad V^{\mathcal{L}} = N_{\text{mid}}^{\mathcal{L}} \otimes^{c_{\text{mid}}} W^V \tag{3}$$

In this formulation, $W^Q$, $W^K$, and $W^V$ denote the weight matrices. The $\otimes^{c_{\text{mid}}}$ operation is analogous to the hyperbolic linear transformations described in equations 1 and 2. Attention weights are derived by computing the Lorentzian inner product between $Q^{\mathcal{L}}$ and $K^{\mathcal{L}}$, then normalized using a Softmax function, and subsequently used for a weighted summation of $V^{\mathcal{L}}$. This entire process is delineated as follows:

$$\alpha_{ij} = \text{Softmax}(\langle (Q^{\mathcal{L}})^T, (K^{\mathcal{L}})^T \rangle_{\mathbb{L}}) \tag{4}$$

$$\text{Att}_i \odot^{c_{\text{mid}}} V_j^{\mathcal{L}} := \frac{\sum_{j=1}^{N} \alpha_{ij} V_j^{\mathcal{L}}}{\sqrt{c_{\text{mid}} \left| \|\sum_{k=1}^{N} \alpha_{ik} V_k^{\mathcal{L}}\|_2^2 \right|}} \tag{5}$$

Here, $\odot^{c_{\text{mid}}}$ signifies the weighted sum in the hyperbolic space, and $\text{Att}_i$ represents the $i$-th row of the attention matrix within the Lorentz model. After the Lorentz centroid operation, the outputs from all attention heads are concatenated along the feature dimension. A subsequent hyperbolic linear transformation then yields the final feature representation $M_{mid}^{\mathcal{L}} \in \mathbb{L}^d, \mathbb{R}^{t \times (d+1)}$.

**Lorentz Residual Connection.** To facilitate deeper features learning within the model, we incorporate a residual connection (He et al., 2025) between the output of the hyperbolic multi-head self-attention mechanism and the original fMRI embedding $F_{mid}^{\mathcal{L}}$. It is formulated as:

$$R_{\text{mid}}^{\mathcal{L}} = \frac{F_{\text{mid}}^{\mathcal{L}} + \beta M_{\text{mid}}^{\mathcal{L}}}{\sqrt{c_{\text{mid}} \left| \langle F_{\text{mid}}^{\mathcal{L}} + \beta M_{\text{mid}}^{\mathcal{L}}, F_{\text{mid}}^{\mathcal{L}} + \beta M_{\text{mid}}^{\mathcal{L}} \rangle_{\mathbb{L}} \right|}} \tag{6}$$

Here, $\beta$ balances the weights of the two feature sets.

**Lorentz MLP.** We introduce a Lorentz MLP network to enhance the model's expressive power. This network integrates non-linearity through intrinsic Lorentz linear layers and activation functions (Yang et al., 2024), which are crucial for extracting richer, higher-level semantic information from the input feature $R_{mid}^{\mathcal{L}}$. Subsequently, the output of the MLP layer is residually connected with $R_{mid}^{\mathcal{L}}$, yielding the universal fMRI embedding $U_{mid}^{\mathcal{L}} \in \mathbb{L}^d, \mathbb{R}^{t \times (d+1)}$, which provides a more robust representation for cross-modal learning.

## 2.2 MULTI-MODAL HYPERBOLIC REPRESENTATION ALIGNMENT

VLMs leverage large-scale multi-modal learning to extract rich visual semantic features within a shared semantic space (Ghosh et al., 2024). These features not only effectively predict neural responses in the human higher-order visual cortex, but their hierarchical processing mechanisms also align closely with the visual information processing in the human brain (Wang et al., 2023; Subramaniam et al., 2024). Inspired by this, we align visual features extracted from a frozen VLM with their corresponding fMRI neural representations in a hyperbolic space. Our objective is to unify image and fMRI embeddings within this shared space, uncovering hierarchical vision-brain relationships via hyperbolic contrastive and entailment losses.

**Lorentzian Centroid Calculation.** Given the extracted image features $I_{mid}^{\mathcal{E}} \in \mathbb{R}^{t \times (p+1)}$, we first transform them from Euclidean geometry to hyperbolic geometry using a Lorentz linear layer, resulting in $I_{mid}^{\mathcal{L}} \in \mathbb{L}^d, \mathbb{R}^{t \times (d+1)}$. To extract more representative features for each sample, we then compute the Lorentz centroids for both the universal fMRI embeddings $U_{mid}^{\mathcal{L}}$ and the image embeddings $I_{mid}^{\mathcal{L}}$. The specific process is described as follows:

$$I_{\text{mid}}^{\mathcal{LC}} = \frac{\sum_{j=1}^{N} I_j^{\mathcal{L}}}{\sqrt{c_{\text{mid}} \left| \left\| \sum_{k=1}^{N} I_k^{\mathcal{L}} \right\|_2^2 \right|}}, \quad U_{\text{mid}}^{\mathcal{LC}} = \frac{\sum_{j=1}^{N} U_j^{\mathcal{L}}}{\sqrt{c_{\text{mid}} \left| \left\| \sum_{k=1}^{N} U_k^{\mathcal{L}} \right\|_2^2 \right|}} \tag{7}$$

Here, both $I_{\text{mid}}^{\mathcal{LC}}$ and $U_{\text{mid}}^{\mathcal{LC}}$ have a dimensionality of $(d+1)$.

**Hyperbolic Contrastive Learning.** In cross-modal learning, aligning and comprehending the relationships between diverse modalities frequently employs the contrastive learning paradigm (Radford et al., 2021). This study leverages hyperbolic embeddings to align visual data with corresponding brain activity. Given a batch of $N$ samples, we utilize the negative Lorentzian distance as a similarity metric to compute the contrastive loss for image-fMRI data pairs. This optimization objective is formally defined by integrating a learnable temperature parameter $\tau$ and the Softmax function:

$$L_{\text{con}}(I, U) = -\sum_{i \in N} \log \frac{\exp\left(-\frac{d_{\mathbb{L}}(I_i^{\mathcal{LC}}, U_i^{\mathcal{LC}})}{\tau}\right)}{\sum_{k=1, k \neq i}^{N} \exp\left(-\frac{d_{\mathbb{L}}(I_i^{\mathcal{LC}}, U_k^{\mathcal{LC}})}{\tau}\right)} \tag{8}$$

Here, for a given image embedding $I_i^{\mathcal{LC}}$, its negative samples are chosen from other fMRI embeddings $U_k^{\mathcal{LC}}(k \neq i)$ within the same batch. Conversely, if an fMRI hyperbolic embedding is the anchor, the loss for negative samples from the batch's image features is defined as $L_{\text{con}}(U, I)$. The hyperbolic contrastive loss, which integrates this bidirectional contrastive process, is formulated as:

$$L_{\text{con}} = \frac{1}{2}(L_{\text{con}}(I, U) + L_{\text{con}}(U, I)) \tag{9}$$

This objective promotes the convergence of matched image and fMRI features on the Lorentz manifold, while separating mismatched ones, achieving effective cross-modal alignment.

**Hyperbolic Entailment Learning.** In addition to the contrastive loss, we introduce an entailment loss to reinforce the partial order relationship between different embeddings (Vendrov et al., 2015). The image embedding represents the model's generalized understanding of an image's core semantics, whereas the fMRI embedding captures more specific neural activity patterns. As depicted in Figure 5 in the Appendix B, an entailment cone (Ganea et al., 2018) is defined for the image embedding $I_{\text{mid}}^{\mathcal{LC}}$ ($I_{cat}$) within the hyperbolic space (i.e., the yellow region). Any fMRI embedding $U_{\text{mid}}^{\mathcal{LC}}$ ($F_{cat}$) falling within this region represents more specific information, indicating its ability to explain

or predict these particular image stimuli. The aperture angle (Le et al., 2019; Desai et al., 2023) of the conical region is defined as follows:

$$\text{aper}(I_{\text{mid}}^{\mathcal{LC}}) = \sin^{-1}\left(\frac{2K}{\sqrt{c_{\text{mid}}}\|I_{\text{mid-space}}^{\mathcal{LC}}\|}\right) \tag{10}$$

Here, the constant $K = 0.1$ provides stable boundary conditions near the origin.

To learn the partial order relationship within this space, specific concepts must reside inside the entailment cone defined by more general concepts. For this purpose, we introduce the entailment loss (Le et al., 2019; Desai et al., 2023), which encourages any fMRI embedding located outside the entailment cone to move towards the boundary of the region delimited by the image embedding:

$$L_{\text{ent}}(I_{\text{mid}}^{\mathcal{LC}}, U_{\text{mid}}^{\mathcal{LC}}) = \max(0, \text{ext}(I_{\text{mid}}^{\mathcal{LC}}, U_{\text{mid}}^{\mathcal{LC}}) - \text{aper}(I_{\text{mid}}^{\mathcal{LC}})) \tag{11}$$

Where $\text{ext}(x, y)$ denotes the external angle of point $y$ with respect to $x$, calculated as:

$$\text{ext}(I_{\text{mid}}^{\mathcal{LC}}, U_{\text{mid}}^{\mathcal{LC}}) = \cos^{-1}\left(\frac{U_{\text{mid-time}}^{\mathcal{LC}} + I_{\text{mid-time}}^{\mathcal{LC}} c_{\text{mid}}\langle I_{\text{mid}}^{\mathcal{LC}}, U_{\text{mid}}^{\mathcal{LC}}\rangle_{\mathbb{L}}}{\|I_{\text{mid-space}}^{\mathcal{LC}}\|\sqrt{(c_{\text{mid}}\langle I_{\text{mid}}^{\mathcal{LC}}, U_{\text{mid}}^{\mathcal{LC}}\rangle_{\mathbb{L}})^2 - 1}}\right) \tag{12}$$

The total loss for our model is a weighted sum of $L_{\text{con}}$ and $L_{\text{ent}}$:

$$L = L_{\text{con}} + \lambda L_{\text{ent}} \tag{13}$$

**Lorentz Classifier.** Furthermore, the aggregated neural representation $U_{\text{mid}}^{\mathcal{LC}}$ guides multi-label prediction through a Lorentz multinomial logistic regression (MLR) layer (Bdeir et al., 2023). This classifier computes the signed hyperbolic distance from input features to learned hyperplanes on the Lorentz manifold, yielding logits for each category via the formula 22 presented in the Appendix B.

## 3 EXPERIMENTS

### 3.1 IMPLEMENTATION DETAILS

We conduct experiments on the NSD (Allen et al., 2022), applying uniform preprocessing to fMRI signals across subjects (S1, S2, S5, S7). HypBrain's performance is compared against SOTA methods on various downstream tasks and evaluated in different geometric spaces. The VLMs employed in our experiments include CLIP (Radford et al., 2021), BLIP-2 (Li et al., 2023), and DeepSeek-Janus-Pro (Chen et al., 2025b). Comprehensive details regarding the dataset, experimental setups, training parameters, and model configurations are provided in Appendix C.

### 3.2 MULTI-LABEL PREDICTION

Multi-label prediction aims to decode the brain's semantic representation of specific visual concepts in observed images. Table 1 shows our method consistently outperforms existing cross-subject decoding approaches (Zhou et al., 2024; Chehab et al., 2022) on all metrics. Notably, HypBrain achieves a substantial improvement over the CLIP-MUSED model (Zhou et al., 2024), with a 10.6% increase in mean Average Precision (mAP) and a 5.2% increase in the area under the receiver operating characteristic curve (AUC). The findings demonstrate HypBrain's capability to both mitigate inter-subject variability and yield hyperbolic features that enhance classification discriminability. As shown in Table 1, the consistently diminished prediction performance of the Euclidean-based model clearly demonstrates the superiority of hyperbolic representations for this task. Qualitative diagrams and all subjects results are presented in Appendix D.1.

### 3.3 BRAIN-IMAGE RETRIEVAL

The retrieval evaluation assesses the amount of image-specific information captured within brain embeddings. We conduct two experiments: image retrieval, which uses a brain embedding to retrieve the most similar image embedding, and brain retrieval, the reverse process. As shown in Table 2, our proposed method achieves highly competitive results against various SOTA models, including

Table 1: Quantitative comparison of multi-label prediction performance against SOTA methods and across embedding spaces, averaged across 4 subjects. Bold font signifies the best performance.

| Manifold | Method | mAP ↑ | AUC ↑ | Hamming ↓ |
|---|---|---|---|---|
| Euclidean | SMODEL-CNN | 0.150 | 0.767 | 0.039 |
| | SMODEL-ViT | 0.156 | 0.755 | 0.038 |
| | EMB | 0.220 | 0.825 | 0.035 |
| | CLIP-MUSED | 0.258 | 0.877 | 0.030 |
| Euclidean | EucBrain | 0.279 | 0.917 | 0.027 |
| Hyperbolic | HypBrain | **0.364** | **0.929** | **0.026** |

**subject-specific models** like MindReader (Lin et al., 2022), BrainDiffuser (Ozcelik & VanRullen, 2023), MindEye (Scotti et al., 2023), and Lite-Mind (Gong et al., 2024). Our cross-subject approach, HypBrain, achieves image and brain retrieval accuracies of 87.8% and 87.4%, respectively. Compared to MindReader (Lin et al., 2022), HypBrain-DeepSeek achieves improvements of 76.8% and 38.4% in retrieval accuracy for both retrieval modalities. This suggests that hyperbolic space is more effective at capturing the inherent hierarchical structures and complex relationships within fMRI and image data, thereby reducing the semantic gap between different modalities. While the MindEye model (Scotti et al., 2023) achieves superior retrieval performance, it relies on a large diffusion model (Ramesh et al., 2022) to learn a conditional distribution from fMRI to image embeddings for each subject. This approach is more akin to data generation that fits low-level image details rather than pure feature alignment, and thus comes with significant computational costs and data dependency (1,003.64M parameters). A crucial finding is that without this generative prior, the MindEye model's performance drops significantly to 88.8% in image retrieval and 84.9% in brain retrieval (MindEye (Backbone + Projector)). In contrast, our HypBrain model, which uses a lightweight hyperbolic encoder and a frozen VLM, achieves precise semantic alignment by solely optimizing the geometric properties of the embedding space. This approach is uniquely positioned to capture high-level, abstract semantic hierarchies and their containment relationships. We argue that for vision-brain research, modeling these high-level hierarchies is more valuable than simply reproducing pixel-level details, offering a more interpretable and scalable perspective for the field.

We further investigate the performance of different embedding spaces in the retrieval task (Table 2). Using cosine similarity in Euclidean space, we find that hyperbolic embedding space significantly outperforms Euclidean space in retrieval accuracy. Specifically, HypBrain-BLIP-2 achieves image retrieval accuracy of 85.3% and brain retrieval accuracy of 84.9%, marking improvements of 2.4% and 13.9% over EucBrain-BLIP-2. This suggests that the latent hierarchical structural properties within the data receive more discriminative embedded representations in hyperbolic geometry. Notably, the advantages of hyperbolic space extend to all three HypBrain variants, underscoring the broad applicability of our framework. Detailed results are available in Appendix D.2.

### 3.4 ABLATION STUDIES

In this section, we conduct ablation studies to further analyze and validate the components of our proposed method. The VLM used in the experiment is DeepSeek-Janus-Pro (Chen et al., 2025b).

**Efficacy of Architectural Design.** We first investigate the impact of fMRI encoder architectures on model performance. As shown in Table 3, our method consistently outperforms the MLP-based architecture (comprising Lorentz MLP and Lorentz residual connection) in both multi-label prediction and retrieval tasks. This demonstrates that the Lorentz Transformer more adeptly captures complex nonlinear relationships within hyperbolic fMRI data, enhancing the model's expressive power.

**Effectiveness of the Entailment Loss.** We analyze the effectiveness of the entailment loss in guiding model learning, as detailed in Table 3. Removing the entailment loss (denoted as w/o $L_{ent}$) leads to a drop in image retrieval accuracy from 87.8% (for our HypBrain model) to 87.1%, indicating its beneficial role in vision-brain representation learning. The complete model consistently achieves superior performance across both metrics, emphasizing the necessity of synergistic contributions from multiple loss functions to enhance model performance.

Table 2: Comprehensive retrieval performance comparison of SOTA methods, HypBrain, and EucBrain variants across embedding spaces. All results are averaged across 4 subjects, expect MindReader (Lin et al., 2022) and MindEye (Backbone + Projector) (Scotti et al., 2023), which only analyzed S1 because the original text only provided results for Subject 1. Bold font signifies the best performance. * denotes a subject-specific model.

| Manifold | Method | Parameters | Image ↑ | Brain ↑ |
|---|---|---|---|---|
| Euclidean | MindReader* | 2.34M | 11.0% | 49.0% |
| | BrainDiffuser* | 4.5B | 21.1% | 30.3% |
| | MindEye* | 1,003.64M | **93.6%** | 90.1% |
| | MindEye (Backbone + Projector)* | 996M | 88.8% | 84.9% |
| | Lite-Mind* | 12.49M | 87.7% | **91.1%** |
| Euclidean | EucBrain-CLIP | 89.74M | 62.3% | 70.5% |
| | EucBrain-BLIP-2 | 90.13M | 82.9% | 71.0% |
| | EucBrain-DeepSeek | 90.07M | 78.1% | 74.4% |
| Hyperbolic | HypBrain-CLIP | 39.41M | 76.2% | 76.3% |
| | HypBrain-BLIP-2 | 39.60M | 85.3% | 84.9% |
| | HypBrain-DeepSeek | 39.41M | 87.8% | 87.4% |

Table 3: Performance comparison of ablation study. All results are averaged across 4 subjects. Bold font signifies the best performance.

| Method | Multi-label prediction | | | Retrieval | |
|---|---|---|---|---|---|
| | mAP ↑ | AUC ↑ | Hamming ↓ | Image ↑ | Brain ↑ |
| Efficacy of Architectural Design | | | | | |
| MLP | 0.322 | 0.911 | 0.026 | 85.7% | 84.5% |
| Effectiveness of the Entailment Loss | | | | | |
| w/o $L_{ent}$ | - | - | - | 87.1% | 87.3% |
| Impact of Lorentz Curvature | | | | | |
| 1.0 | 0.332 | 0.915 | 0.027 | 87.7% | 87.3% |
| 2.0 | 0.310 | 0.913 | 0.029 | 87.7% | 83.4% |
| 3.0 | 0.248 | 0.893 | 0.029 | 86.7% | 83.2% |
| HypBrain | **0.364** | **0.929** | **0.026** | **87.8%** | **87.4%** |

**Impact of Lorentz Curvature.** In the Lorentz model, the setting of curvature is crucial for effective embedding representation learning. To identify the optimal configuration, we compare the performance impact of a fixed versus a learnable $c_{mid}$ parameter. Table 3 demonstrates that treating curvature as a learnable parameter consistently achieves superior results in both tasks.

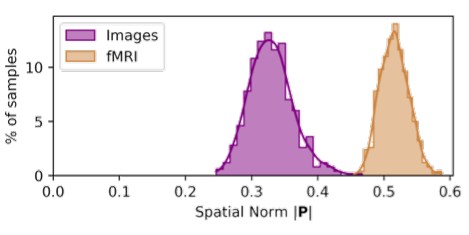
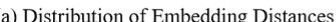
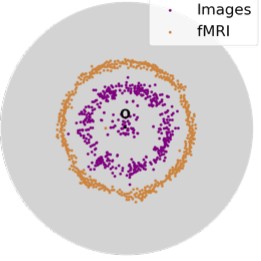

(a) Distribution of Embedding Distances      (b) CO-SNE Visualization

Figure 3: Visualization of the learned hyperbolic space by HypBrain-DeepSeek model. Elements closer to the origin exhibit higher semantic hierarchy and coarser granularity.

### 3.5 VISUALIZATION OF HYPERBOLIC SPACE

To illuminate the semantic distribution of fMRI and image embeddings, we undertake a low-dimensional visualization of the learned hyperbolic space. Specifically, we randomly sample 1K embeddings from the training set and analyze their norm distribution through a histogram. Subsequently, these embeddings are mapped into a low-dimensional space using CO-SNE (Guo et al., 2022) for better observation. Figure 3(a) clearly shows that image embeddings are positioned closer to the center of the hyperbolic space compared to fMRI embeddings. This supports the effectiveness of using $L_{ent}$ to guide the learning of a partial order relationship between brain activity and images. Furthermore, Figure 3(b) reveals a distinct semantic separation and a clear hierarchical structure in the distribution of both image and corresponding fMRI embeddings. This indicates that our method successfully captures semantic associations and hierarchical relationships between different modalities within the hyperbolic space. Results for other models can be found in Appendix D.3.

### 3.6 CROSS-SUBJECT GENERALIZATION ANALYSIS

High-resolution fMRI signal acquisition presents challenges for large-scale data collection due to its time-consuming and labor-intensive nature. To overcome this, we employ a cross-subject training strategy, facilitating effective generalization to new subjects with limited data. We validate this by training the model on data from three subjects (S1, S2, S5) and evaluating its generalization on a new subject (S7) using varying data proportions. Specifically, we train a subject-specific hyperbolic tokenizer for S7, followed by two settings: freezing and fine-tuning the universal hyperbolic encoder. Figure 4 illustrates that the fine-tuning approach achieves performance comparable to the HypBrain-DeepSeek model with only 50% of the data. This highlights the model's ability to learn from other subjects and integrate minimal new subject information, effectively addressing data scarcity and demonstrating strong generalization. Moreover, fine-tuning slightly surpasses the frozen encoder, adapting better to individual differences in new subjects. More results are in Appendix D.4.

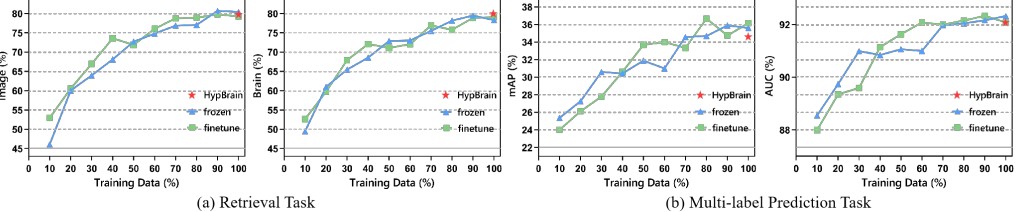

(a) Retrieval Task          (b) Multi-label Prediction Task

Figure 4: Cross-subject generalization analysis across different downstream tasks.

## 4 CONCLUSION

In this paper, we introduce HypBrain, a novel hyperbolic space-based framework for cross-subject vision-brain representation learning. By aligning neural responses and image representations in a shared Lorentz manifold, our method effectively captures cross-modal semantic hierarchical relationships. We design a hyperbolic fMRI encoder that extracts both shared and individual-specific brain patterns. To achieve robust semantic alignment, we incorporate a novel optimization strategy combining hyperbolic contrastive loss and partial order entailment constraints, yielding more discriminative hyperbolic embeddings. Extensive experiments demonstrate HypBrain consistently surpasses Euclidean-based approaches and performs comparably to or surpasses existing SOTA methods. Our visualization analyses further confirm HypBrain's capability to geometrically embed multi-modal hierarchical structures in hyperbolic space. Our work not only opens a new perspective for vision-brain representation learning but also underscores the immense potential of hyperbolic geometry in modeling complex cross-modal semantic relationships. While our research primarily focuses on optimizing the geometric properties of the embedding space for high-level semantic alignment, we acknowledge the potential of integrating generative models to enhance multi-modal mappings. Future work will explore a hybrid framework that leverages the benefits of hyperbolic geometry for hierarchy while also incorporating generative priors for richer, low-level data fidelity.

## REPRODUCIBILITY STATEMENT

Our source code is publicly available at `https://anonymous.4open.science/r/HypBrain-83F1/`. In all experiments, we use public datasets. Further details on the VLM embeddings, training parameters, and model configurations used in the experiments can be found in Appendix C.

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

# APPENDIX

## A    RELATED WORK

**Vision-brain Semantic Alignment.** Learning the semantic relationship between fMRI neural activity and visual stimuli has garnered significant research interest in recent years. Existing studies primarily focus on constructing this semantic alignment within a shared latent space (Scotti et al., 2023; Ozcelik & VanRullen, 2023; Han et al., 2024; Xia et al., 2024). With the advent of VLMs, several approaches map fMRI modalities into pretrained embedding spaces, either through direct regression (Takagi & Nishimoto, 2023a; Ozcelik & VanRullen, 2023; Takagi & Nishimoto, 2023b) or contrastive learning (Xia et al., 2024). MindEye (Scotti et al., 2023), for instance, encodes images using CLIP (Radford et al., 2021) and subsequently projects corresponding fMRI data into the CLIP feature space, achieving robust image retrieval performance. Furthermore, some research (Han et al., 2024) treats brain signals as an emerging modality, learning alignments between multi-modal inputs via generative training. However, these methods typically rely on Euclidean spaces to capture simple semantic similarities, fundamentally overlooking the intricate hierarchical structures inherent in the data. To address this, we conceptualize image data and their corresponding fMRI responses as entities possessing distinct levels of abstraction. We embed these into a unified hyperbolic space for semantic alignment, thereby overcoming the limitations associated with Euclidean geometry.

**Learning in Hyperbolic Space.** Hyperbolic manifolds are gaining increasing attention in deep learning due to their effectiveness in modeling hierarchical structures. Initially, MERU (Desai et al., 2023) combines entailment learning and CLIP methods to learn embeddings in hyperbolic space, thereby capturing underlying vision-language hierarchical relationships. With the advancement of hyperbolic neural networks, recent works apply hyperbolic models to various modalities, including images, text, video, and medical imaging (Srivastava & Wu; Yang et al., 2024; Wang et al., 2024b; Pal et al., 2024; Kwon et al., 2024; Gonzalez-Jimenez et al., 2025; Li et al., 2025). Notably, Alvaro Gonzalez-Jimenez et al. (Gonzalez-Jimenez et al., 2025) introduce a hyperbolic space-based framework for medical anomaly detection and localization, which presents a significant breakthrough in this field. HyperVLM (Srivastava & Wu) further explores hierarchical relationships between images and text through hyperbolic Poincaré geometry properties, establishing a novel contrastive learning paradigm. Furthermore, hyperbolic learning is also integrated into video retrieval tasks, leveraging its hierarchical modeling capabilities to achieve SOTA performance (Li et al., 2025). Despite these breakthroughs, the potential of hyperbolic geometry in learning vision-brain semantic relationships remains largely underexploited. To address this critical gap, we propose HypBrain, a model that captures intricate semantic hierarchical associations between different modalities in hyperbolic space, offering a new perspective for vision-brain representation learning.

**Hyperbolic Geometry and Brain.** Hyperbolic geometry, a non-Euclidean geometry characterized by negative curvature, offers significant advantages in handling hierarchical data (Nickel & Kiela,

2018; Ratcliffe, 2006; Bridson & Haefliger, 2013; Chamberlain et al., 2017). Many cognitive functions of the brain also exhibit clear bottom-up hierarchical structures (Joseph et al., 2024). Taking the visual system as an example, fMRI signals under natural image stimulation clearly reveal the brain's hierarchical processing from pixels to meaning. This process involves an exponential expansion of information, transitioning from local brain region activation for low-level features to distributed network activities representing high-level semantics (Bill et al., 2020; Huff et al., 2018). Given this inherent correspondence, we posit that employing hyperbolic space to describe these cognitive processes is more natural and accurate than traditional Euclidean space. Inspired by this, we explore the effectiveness of hyperbolic latent geometry in modeling the intrinsic hierarchical structure of the brain and visual information in this work.

## B  PRELIMINARIES

In this section, we introduce concepts related to hyperbolic geometry briefly. Unlike Euclidean and spherical spaces, hyperbolic space is characterized as a Riemannian manifold with constant negative curvature. A key property of hyperbolic space is that the volume of its subregions increases exponentially with their radius (Bridson & Haefliger, 2013). This characteristic makes hyperbolic geometry particularly well-suited for studying and representing data that inherently possesses hierarchical or tree-like structures (Krioukov et al., 2010). Among various models for hyperbolic space, we adopt the Lorentz model (Nickel & Kiela, 2018; Mishne et al., 2023) as the foundational framework due to its numerical stability and computational efficiency.

**Lorentz Model.** The Lorentz model defines a d-dimensional Riemannian manifold, denoted as $\mathbb{L}_d$, which is a $d$ dimensional hyperboloid embedded within a $d+1$-dimensional Minkowski space. It is formally described as:

$$\mathbb{L}_d = \left\{ x \in \mathbb{R}^{d+1} \mid \langle x, x \rangle_{\mathbb{L}} = -\frac{1}{c}, x_0 > 0 \right\} \tag{14}$$

Here, $c$ represents the curvature of the space, and $\langle ., . \rangle_{\mathbb{L}}$ is the Lorentzian inner product defined for $x, y \in \mathbb{R}^d$ as:

$$\langle x, y \rangle_{\mathbb{L}} = -x_0 y_0 + \sum_{i=1}^{d} x_i y_i \tag{15}$$

As established in (Chen et al., 2021), the 0-th element of vector $x$ corresponds to the temporal dimension, while the remaining components constitute the spatial dimensions. Consistent with the definition of $\mathbb{L}_d$, the temporal component $x_0$ is determined by its spatial counterpart $x_{space}$, specifically through the relationship:

$$x_{\text{time}} = x_0 = \sqrt{\frac{1}{c} + \|x_{\text{space}}\|^2} \tag{16}$$

Where $\| \cdot \|$ denotes the Euclidean norm and $x_{\text{space}} = x_{1:d}$.

**Tangent Spaces.** A tangent space at a point $x \in \mathbb{L}_d$ is a Euclidean space that is orthogonal to it, defined as:

$$T_x \mathbb{L}^d := \left\{ y \in \mathbb{R}^{d+1} \mid \langle y, x \rangle_{\mathbb{L}} = 0 \right\} \tag{17}$$

The tangent space at the origin $O = \left( \sqrt{\frac{1}{c}}, 0, \ldots, 0 \right)^T$ is denoted as $T_O \mathbb{L}_d$.

**Exponential Map.** To perform operations within the hyperbolic space, we utilize the exponential map, which projects vectors from the tangent space onto the hyperbolic manifold. Specifically, given a vector $a$ in the tangent space $T_x \mathbb{L}^d$, its exponential map is implemented as:

$$\exp_x^c(a) = \cosh(\sqrt{c}\|a\|_{\mathbb{L}})x + \frac{\sinh(\sqrt{c}\|a\|_{\mathbb{L}})}{\sqrt{c}\|a\|_{\mathbb{L}}} a \tag{18}$$

**Logarithmic Map.** Conversely, the logarithmic map facilitates the projection from the Lorentz model back to the tangent space:

$$\log_x^c(y) = \frac{\cosh^{-1}(-c\langle x, y \rangle_{\mathbb{L}})}{\sqrt{(-c\langle x, y \rangle_{\mathbb{L}})^2 - 1}}(y + c\langle x, y \rangle_{\mathbb{L}} x) \tag{19}$$

**Geodesics.** In hyperbolic space, geodesics represent the shortest paths connecting two points, analogous to straight lines in Euclidean geometry. Within the Lorentz model, a geodesic is defined by the intersection of a plane passing through the origin and the hyperboloid. The Lorentzian distance between two points, $x$ and $y$, is calculated as:

$$d_{\mathbb{L}}(x, y) = \frac{1}{\sqrt{c}} \cosh^{-1}(-c\langle x, y\rangle_{\mathbb{L}}) \tag{20}$$

**Lorentzian Centroid.** The Lorentzian centroid is defined in the hyperbolic space $\mathbb{L}^d$ by minimizing the weighted squared Lorentzian distance, which yields the following expression:

$$\mu = \frac{\sum_{i=1}^{m} w_i x_i}{\left|\|\sum_{i=1}^{m} w_i x_i\|_{\mathbb{L}}\right|} \tag{21}$$

Here, $x_i \in \mathbb{L}^d$ denotes a point in the hyperbolic space, and $w_i$ represents its corresponding non-negative weight.

**Lorentz MLR layer.** This appendix provides the detailed computational formula for the Lorentz classifier, which is crucial for multi-label prediction as discussed in the main text. The logits for each category are yielded via the following formula:

$$\text{logits}(U_{\text{mid}}^{\mathcal{LC}}) = \text{sign}(\alpha) \cdot \gamma \cdot \left(\sqrt{c_{\text{out}}} \cdot \left|\sinh^{-1}\left(\frac{1}{\sqrt{c_{\text{out}}}} \cdot \frac{\alpha}{\gamma}\right)\right|\right) \tag{22}$$

Here, $\alpha$ is the projection of $U_{\text{mid}}^{\mathcal{LC}}$ onto the hyperplane, $\gamma$ is the Lorentz norm of the hyperplane's normal vector, and $c_{out}$ corresponds to output manifold curvature. The classifier undergoes training using a cross-entropy loss function.

**Hyperbolic Entailment Learning.** This figure conceptually illustrates the entailment cone (shown in yellow) in hyperbolic space, originating from an image embedding $I_{\text{mid}}^{\mathcal{LC}}$ ($I_{cat}$). It demonstrates how an fMRI embedding $U_{\text{mid}}^{\mathcal{LC}}$ ($F_{cat}$) falling within this cone represents a more specific neural pattern, indicating its hierarchical relationship with the image's generalized semantics. The cone's aperture angle defines the scope of this entailment, supporting our use of an entailment loss.

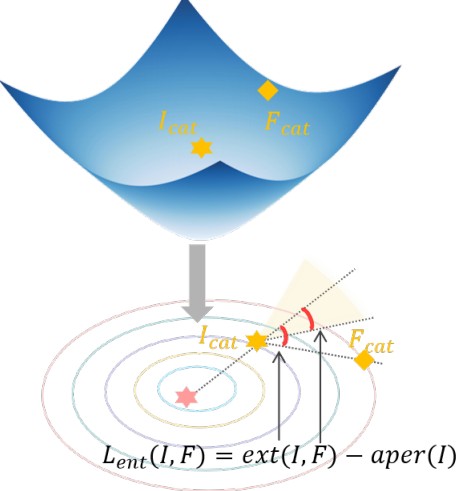

Figure 5: Intuitive examples of entailment loss and related concepts.

## C  EXPERIMENTAL DETAILS

In this section, we provide a comprehensive description of the experimental details. This includes a thorough explanation of the datasets utilized, the specific training configurations employed, and the implementation specifics for all downstream tasks. Furthermore, we detail the corresponding evaluation benchmarks and metrics used to assess performance.

## C.1 DATASET AND PROCESSING

The Natural Scenes Dataset (NSD) (Allen et al., 2022) stands as the most extensive publicly accessible functional magnetic resonance imaging (fMRI) dataset, encompassing high-resolution 7-Tesla fMRI scans acquired from eight participants. However, this study utilizes data exclusively from a subset of four participants (S1, S2, S5, S7). For the training set, each participant contributes 8,859 image stimuli and 24,980 fMRI trials. Conversely, the test set for each participant comprises 982 image stimuli and 2,770 fMRI trials. A crucial distinction is that the image stimuli in the training set are unique to each participant, whereas the image stimuli in the test set are consistent across all participants. Furthermore, owing to inherent variations in brain size and structure, the number of voxels within regions of interest (ROIs) exhibits variability across participants, typically ranging from approximately 13,000 to 16,000 voxels per participant. To facilitate a standardized analysis, these fMRI signals undergo subsequent upsampling, ensuring a uniform voxel count of 18,000 for all participants. Within the NSD, all images employed during fMRI recording originate from the COCO dataset (Lin et al., 2014). The COCO dataset comprises a total of 80 distinct categories. Each image viewed by participants contains multiple COCO labels. Specific category and label information is detailed in Table 4.

Table 4: Statistical information of elements for 'primary categories' and 'multi labels' in NSD.

| Category | COCO Label |
| --- | --- |
| person | person |
| vehicle | bicycle, car, motorcycle, airplane, bus, train, truck, boat |
| outdoor | traffic light, fire hydrant, stop sign, parking meter, bench |
| animal | bird, cat, dog, horse, sheep, cow, elephant, bear, zebra, giraffe |
| accessory | backpack, umbrella, handbag, tie, suitcase |
| sports | frisbee, skis, snow board, sports ball, kite, baseball bat, baseball glove, skateboard, surfboard, tennis racket |
| kitchen | bottle, wine glass, cup, fork, knife, spoon, bowl |
| food | banana, apple, sandwich, orange, broccoli, carrot, hot dog, pizza, donut, cake |
| furniture | chair, couch, potted plant, bed, dining table, toilet |
| electronics | tv, laptop, mouse, remote, keyboard, cell phone |
| appliance | microwave, oven, toaster, sink, refrigerator |
| indoor | book, clock, vase, scissors, teddy bear, hair drier, toothbrush |

## C.2 BASELINE COMPARISONS

To demonstrate the superiority of Lorentzian representations over Euclidean counterparts, we compare HypBrain's performance against existing SOTA methods in multi-label prediction and retrieval tasks. Simultaneously, by evaluating HypBrain's performance in both geometric spaces, we further elucidate the superiority of hyperbolic representations. The VLMs employed in this study include CLIP (Radford et al., 2021), BLIP-2 (Li et al., 2023), and DeepSeek-Janus-Pro (Chen et al., 2025b).

## C.3 TRAINING CONFIGURATIONS

We employ a customized optimization strategy. Specifically, Euclidean parameters are optimized using the AdamW optimizer (Loshchilov & Hutter, 2017), while hyperbolic parameters are updated with the Riemannian Adam optimizer (Bécigneul & Ganea, 2018). Both parameter types are initialized with a learning rate of 2e-4, which adaptively decays based on validation performance, with a weight decay coefficient of 0.1. Training is conducted on a single NVIDIA GeForce RTX 4090 GPU, utilizing a batch size of 32. The models are trained for 200 epochs for retrieval tasks and 100 epochs for classification tasks. The hyperparameter $\lambda$ is set to 0.01.

## C.4 MODEL CONFIGURATIONS

Our model uses a 512-dimensional Lorentz manifold for hyperbolic space transformation. The number of tokens $t$ for fMRI representations is adjusted according to the visual feature dimensions extracted by the VLMs. Specifically, the image feature sizes extracted from the CLIP (Radford et al.,

2021), BLIP-2 (Li et al., 2023), and DeepSeek-Janus-Pro (Chen et al., 2025b) image encoders are $257 \times 1024$, $257 \times 1408$. and $576 \times 1024$, respectively. The learnable curvature parameters, $c_{mid}$ and $c_{out}$, are initialized to 1 and 2, respectively, and are constrained within the range [0.1,10] to prevent training instability. Conversely, $c_{in}$ is a fixed curvature parameter with a value of 1. The temperature parameter $\tau$ is initialized to 0.07 and clamped at 0.01. All these scalar values are learned in logarithmic space.

### C.5 EVALUATION METRIC

#### C.5.1 MULTI-LABEL PREDICTION

The multi-label semantic classification task seeks to simulate the brain's coarse-to-fine multi-level semantic understanding in visual cognition, leading to richer and more accurate image descriptions. We employ three commonly used evaluation metrics in the field of multi-label classification: mean Average Precision (mAP), the area under the receiver operating characteristic curve (AUC) and Hamming distance.

**mAP.** mAP evaluates the average precision across all labels in a multi-label classification task. It reflects the model's overall performance in ranking predictions correctly.

**AUC.** AUC measures the model's ability to discriminate between classes for each label in both binary and multi-label classification tasks. A higher value, approaching 1, indicates better performance.

#### C.5.2 BRAIN-IMAGE RETRIEVAL

Retrieval is to search for pertinent results in response to a provided query from a large database, often considered as a form of fine-grained, instance-level classification. We adopt accuracy as the primary evaluation metric. For each test sample, we randomly select 299 images from the remaining 981 images in the test set and calculate the negative Lorentz distance between the voxel embeddings and 300 images. The retrieval accuracy refers to the proportion of successful retrieval of corresponding images in the 982 voxel embeddings of the test set. We adjust the random number seed of 30 randomly selected images to average the accuracy of all samples.

## D   ADDITIONAL EXPERIMENTS

This section comprehensively demonstrates the superiority of HypBrain through additional experiments, presenting both quantitative and qualitative results.

### D.1 MULTI-LABEL PREDICTION

Figure 6 presents the qualitative prediction results of HypBrain. The model exhibits varying performance when predicting different labels for Subject 1 (S1). Specifically, labels with relatively higher frequencies in visual responses, such as "person", "clock", and "sink" achieve high average precision scores of 96.2%, 51.6%, and 74.0%, respectively. Conversely, labels with lower frequencies, including "toaster", "hair dryer", and "parking meter" show comparatively lower average precision scores of 3.6%, 2.5%, and 1.7%. Furthermore, Table 5 provides a detailed overview of the classification results for all subjects (S1, S2, S5, and S7) across different geometric spaces, highlighting the significant superiority of the hyperbolic representation space. Figure 7 additionally illustrates the results of three evaluation metrics for different subjects using the HypBrain method. It is evident that different subjects demonstrate high consistency in multi-label prediction capabilities, with a difference in AUC scores of less than 1.8%. This further indicates the robust and generalizable nature of our model for multi-label prediction tasks.

### D.2 BRAIN-IMAGE RETRIEVAL

To evaluate the performance of the HypBrain framework in retrieval tasks, we conduct experiments on four subjects (S1, S2, S5, and S7), and the results are presented in Table 6. We compare several variants of HypBrain with two prominent methods: MindEye (Scotti et al., 2023), and Lite-

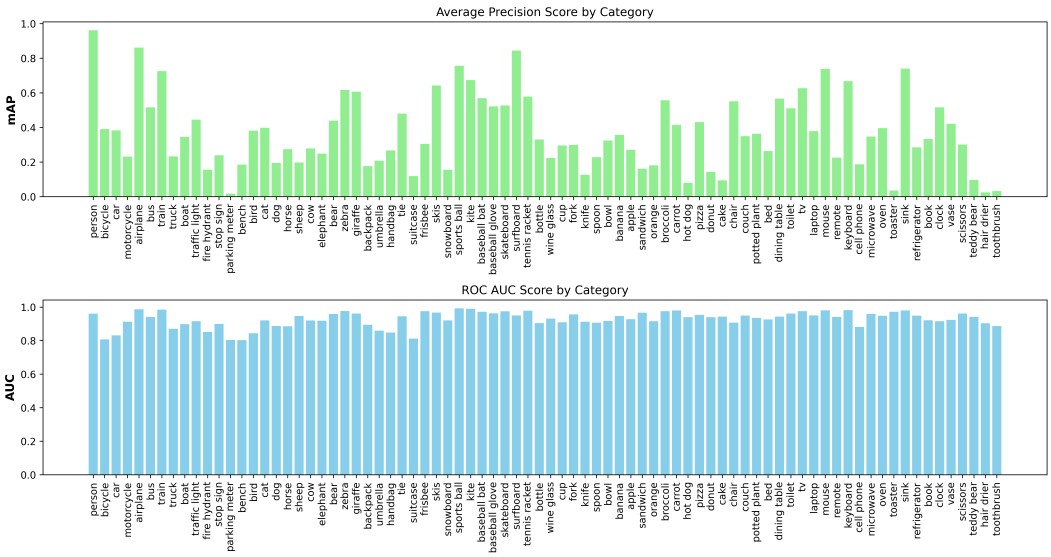

Figure 6: Multi-label prediction metrics for Subject 1 across 80 labels from the NSD.

Table 5: Quantitative comparison of HypBrain multi-label prediction performance across all subjects and embedding spaces.

| Manifold | Subject | mAP ↑ | AUC ↑ | Hamming ↓ |
|---|---|---|---|---|
| Euclidean | S1 | 0.278 | 0.916 | 0.026 |
| | S2 | 0.268 | 0.916 | 0.027 |
| | S5 | 0.305 | 0.926 | 0.026 |
| | S7 | 0.263 | 0.912 | 0.027 |
| Hyperbolic | S1 | 0.370 | 0.928 | 0.026 |
| | S2 | 0.345 | 0.925 | 0.026 |
| | S5 | 0.398 | 0.939 | 0.026 |
| | S7 | 0.342 | 0.921 | 0.027 |

Mind (Gong et al., 2024). The experimental results demonstrate that the HypBrain method exhibits competitive performance in both image and brain retrieval accuracy across different subjects. To further investigate the influence of different embedding spaces on retrieval performance, Table 7 illustrates the retrieval performance of various subjects within distinct embedding spaces. The results consistently indicate the superiority of hyperbolic geometry over Euclidean space for these tasks. Importantly, the consistent strong performance across different HypBrain variants underscores the versatility and robustness of our proposed framework.

Figure 8 illustrates the retrieval results for both MindEye (Scotti et al., 2023) and our proposed HypBrain-DeepSeek method on Subject 5 (S5). The upper panel displays the reference image along with the Top-5 retrieved images from the image retrieval task, while the lower panel presents the Top-5 results for brain retrieval. Furthermore, Figure 9 details the Top-1 retrieval performance across different subjects for both retrieval tasks. Consistent with our quantitative analysis, subjects S1, S2, and S5 achieve high Top-1 accuracy in both image and brain retrieval. Although Subject S7 exhibits slightly lower performance compared to other subjects, this does not diminish the strong generalization capabilities demonstrated by our method.

## D.3 VISUALIZATION OF HYPERBOLIC SPACE

We present low-dimensional visualizations of the hyperbolic spaces learned by two additional Hyp-Brain model variants: HypBrain-CLIP and HypBrain-BLIP-2. These visualizations are depicted in

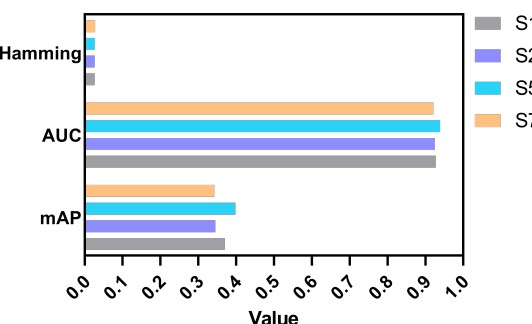

Figure 7: Comparison of multi-label prediction performance of the HypBrain method across different subjects.

Table 6: Quantitative comparison of the retrieval performance of HypBrain against SOTA methods across all subjects. * denotes a subject-specific model.

| Method | Subject | Image ↑ | Brain ↑ |
|---|---|---|---|
| MindEye* | | 97.2% | 94.7% |
| MindEye (Backbone + Projector)* | | 88.8% | 84.9% |
| Lite-Mind* | S1 | 94.6% | 97.4% |
| HypBrain-CLIP | | 78.7% | 79.6% |
| HypBrain-BLIP-2 | | 88.1% | 85.8% |
| HypBrain-DeepSeek | | 90.2% | 89.0% |
| MindEye* | | 97.1% | 93.9% |
| Lite-Mind* | | 94.1% | 98.2% |
| HypBrain-CLIP | S2 | 76.8% | 74.0% |
| HypBrain-BLIP-2 | | 84.4% | 85.5% |
| HypBrain-DeepSeek | | 89.4% | 88.4% |
| MindEye* | | 90.7% | 85.7% |
| Lite-Mind* | | 80.5% | 86.3% |
| HypBrain-CLIP | S5 | 80.5% | 81.5% |
| HypBrain-BLIP-2 | | 89.5% | 89.3% |
| HypBrain-DeepSeek | | 91.5% | 92.7% |
| MindEye* | | 89.4% | 85.9% |
| Lite-Mind* | | 81.7% | 82.3% |
| HypBrain-CLIP | S7 | 69.7% | 69.9% |
| HypBrain-BLIP-2 | | 79.2% | 79.2% |
| HypBrain-DeepSeek | | 80.2% | 79.7% |

Figure 10. Consistent with our observations in Figure 3, the image embeddings in both variants tend to cluster closer to the center of the hyperbolic space, while the fMRI embeddings are distributed in more peripheral regions. This pattern suggests that the effective learning of partial order relationships among different concepts remains robust, even when alternative pretrained models are employed for extracting image features. Furthermore, both HypBrain-CLIP and HypBrain-BLIP-2 demonstrate clear semantic separation and hierarchical structures within their respective image and fMRI embedding spaces. This consistency underscores the generalizability of our proposed method in capturing semantic associations across diverse multi-modal data.

## D.4 CROSS-SUBJECT GENERALIZATION ANALYSIS

A notable advantage of the HypBrain method is its ability to generalize effectively to unseen subjects. This means our model can learn efficiently and perform well even with limited data from entirely new participants. This capability is crucial given the inherent challenges and high costs

Table 7: Retrieval performance comparison across Euclidean (EucBrain-) and Hyperbolic (HypBrain-) embedding spaces and methods for all subjects.

| Manifold | Method | Subject | Image ↑ | Brain ↑ |
|---|---|---|---|---|
| Euclidean | EucBrain-CLIP
EucBrain-BLIP-2
EucBrain-DeepSeek | S1 | 70.6%
87.0%
83.1% | 77.4%
77.0%
81.3% |
| Hyperbolic | HypBrain-CLIP
HypBrain-BLIP-2
HypBrain-DeepSeek | S1 | 78.7%
88.1%
90.2% | 79.6%
85.8%
89.0% |
| Euclidean | EucBrain-CLIP
EucBrain-BLIP-2
EucBrain-DeepSeek | S2 | 65.6%
86.1%
81.6% | 74.1%
72.8%
76.4% |
| Hyperbolic | HypBrain-CLIP
HypBrain-BLIP-2
HypBrain-DeepSeek | S2 | 76.8%
84.4%
89.4% | 74.0%
85.5%
88.4% |
| Euclidean | EucBrain-CLIP
EucBrain-BLIP-2
EucBrain-DeepSeek | S5 | 57.3%
82.7%
78.1% | 67.2%
69.9%
72.6% |
| Hyperbolic | HypBrain-CLIP
HypBrain-BLIP-2
HypBrain-DeepSeek | S5 | 80.5%
89.5%
91.5% | 81.5%
89.3%
92.7% |
| Euclidean | EucBrain-CLIP
EucBrain-BLIP-2
EucBrain-DeepSeek | S7 | 55.6%
75.8%
69.6% | 63.1%
64.3%
67.1% |
| Hyperbolic | HypBrain-CLIP
HypBrain-BLIP-2
HypBrain-DeepSeek | S7 | 69.7%
79.2%
80.2% | 69.9%
79.2%
79.7% |

associated with fMRI data acquisition, as it significantly reduces the reliance on extensive data sampling from new subjects. To validate this generalization ability, we employ a strategy similar to the main experiments. We first pretrained the model on fMRI datasets from three subjects to learn universal brain activity patterns. Subsequently, we fine-tuned this pretrained model using varying proportions of data (ranging from 10% to 100%) from other subjects. This approach allows us to observe the model's generalization performance at the individual subject level. Figure 11 and Figure 12 visually illustrate the model's cross-subject generalization capabilities. They detail the model's performance under two conditions: one where the universal hyperbolic encoder is frozen, and another where it is fine-tuned. Consistent with the generalization observed for subject S7 in the main text, our tests on subjects S1, S2, and S5 also demonstrate that the fine-tuning approach achieves performance comparable to the HypBrain-DeepSeek model trained with the complete dataset, using only 50% of the data. This outcome further underscores the significant superiority of our cross-subject strategy, indicating that the HypBrain model not only learns effectively from limited data but also maintains strong generalization ability and robustness across different individuals.

## E  THE USE OF LARGE LANGUAGE MODELS (LLMs)

During the preparation of this paper, a Large Language Model (LLM) is utilized as a general-purpose assist tool, primarily for polishing and optimizing the writing. The LLM assists in improving the text's grammar, spelling, sentence structure, and overall fluency.

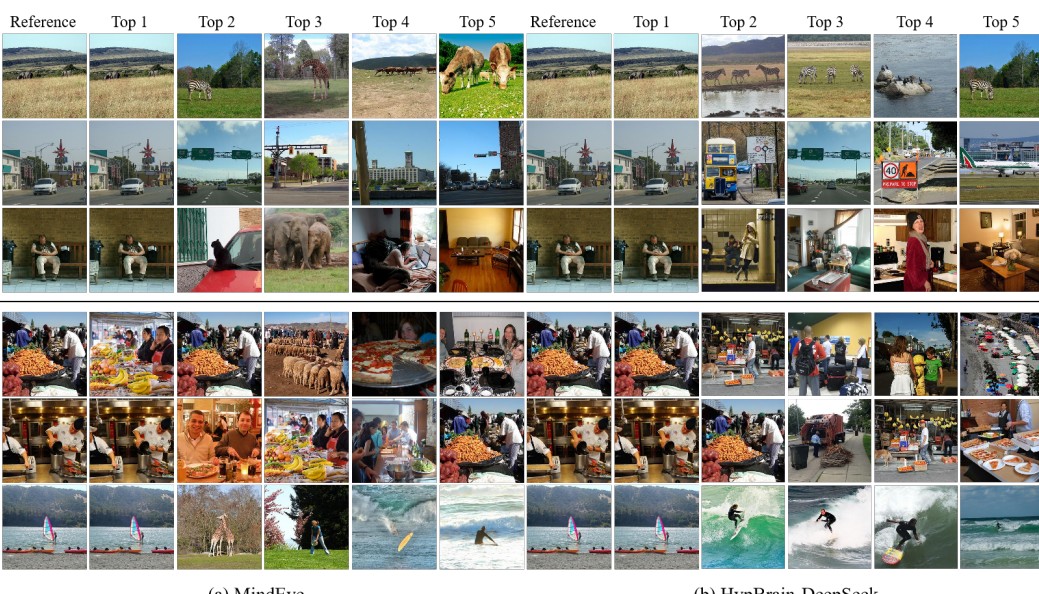

(a) MindEye        (b) HypBrain-DeepSeek

Figure 8: Retrieval examples of (a) MindEye (Scotti et al., 2023) and (b) HypBrain-DeepSeek from NSD for Subject 5. The image retrieval (top) is to find the correct image embedding given a brain embedding. Conversely, the brain retrieval (bottom) aims to locate the correct brain embedding given an image embedding.

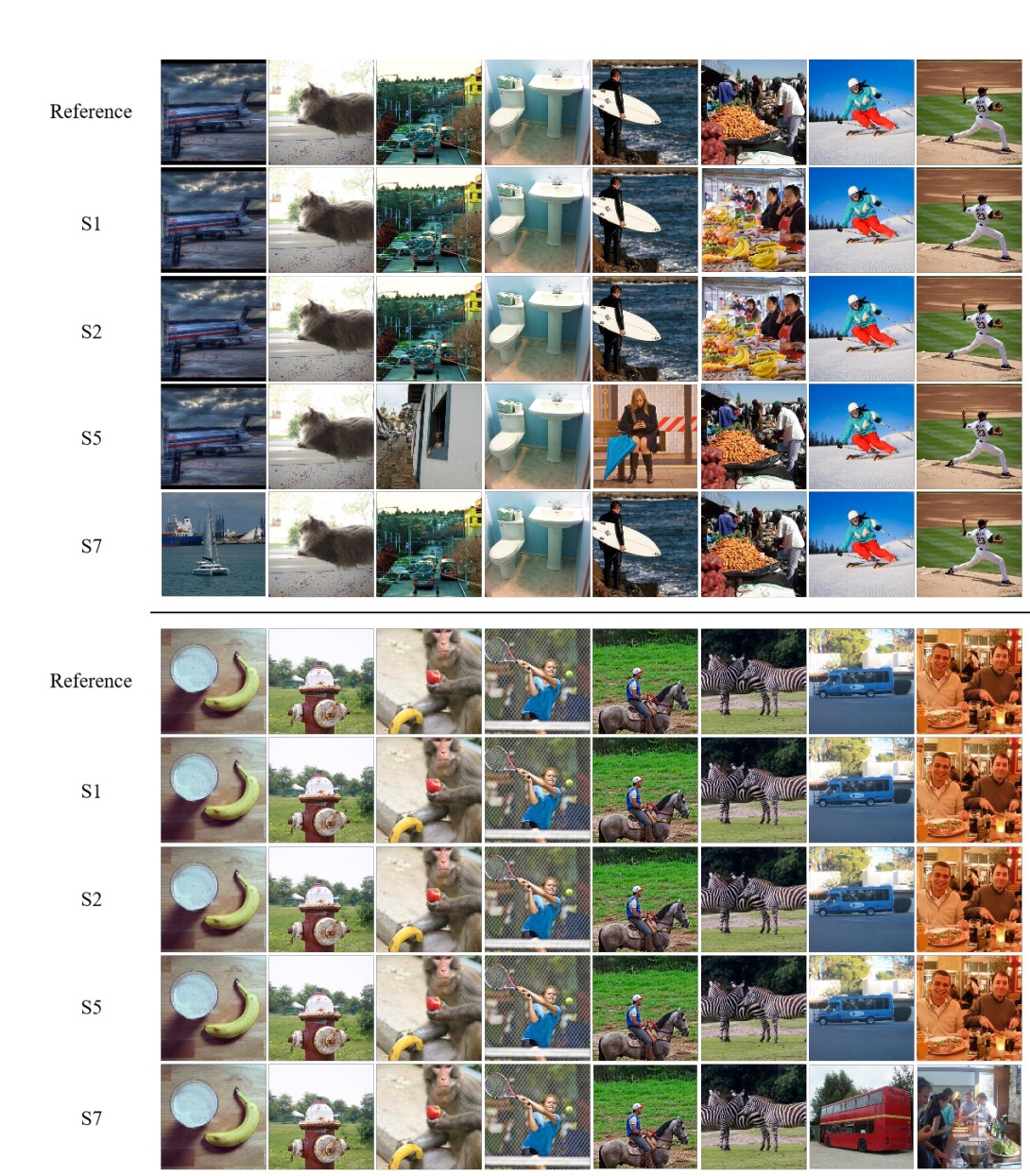

Figure 9: Sample Top-1 retrieval results for different subjects using the cross-subject HypBrain-DeepSeek method. The top section of the figure displays image retrieval results, while the bottom section shows brain retrieval results.

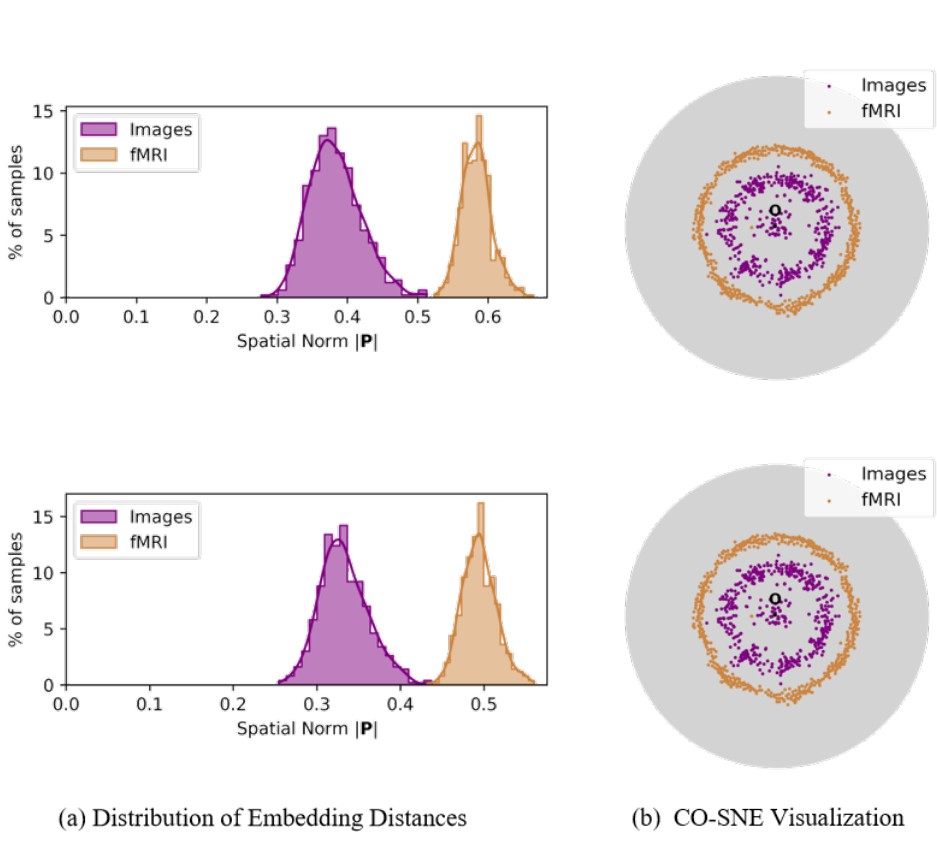

(a) Distribution of Embedding Distances          (b) CO-SNE Visualization

Figure 10: Visualization of the learned hyperbolic spaces across various HypBrain model variants. The top row represents the HypBrain-CLIP model, and the bottom row represents the HypBrain-BLIP-2 model.

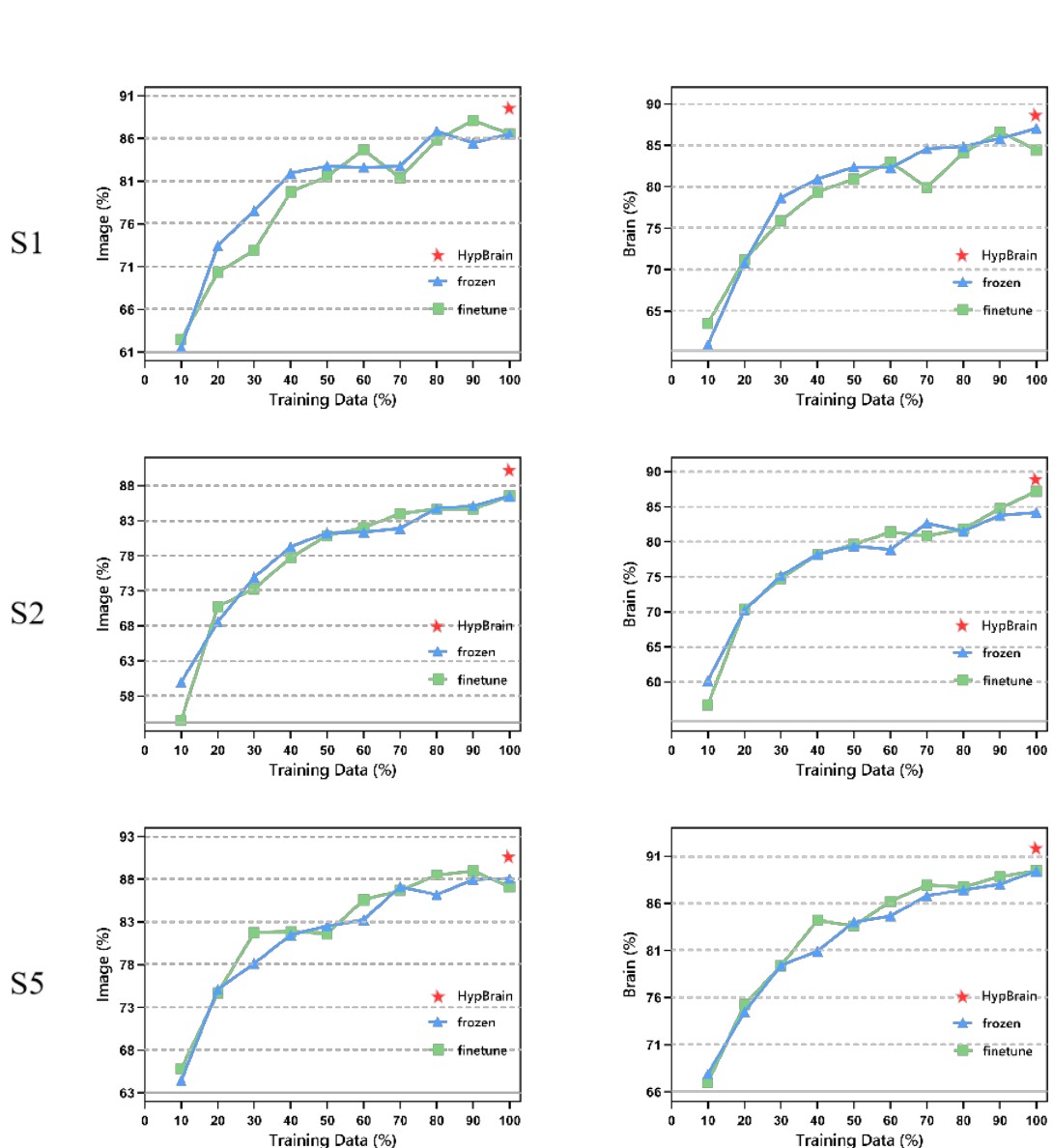

Figure 11: Cross-subject generalization of HypBrain-DeepSeek on unseen subjects in the retrieval task.

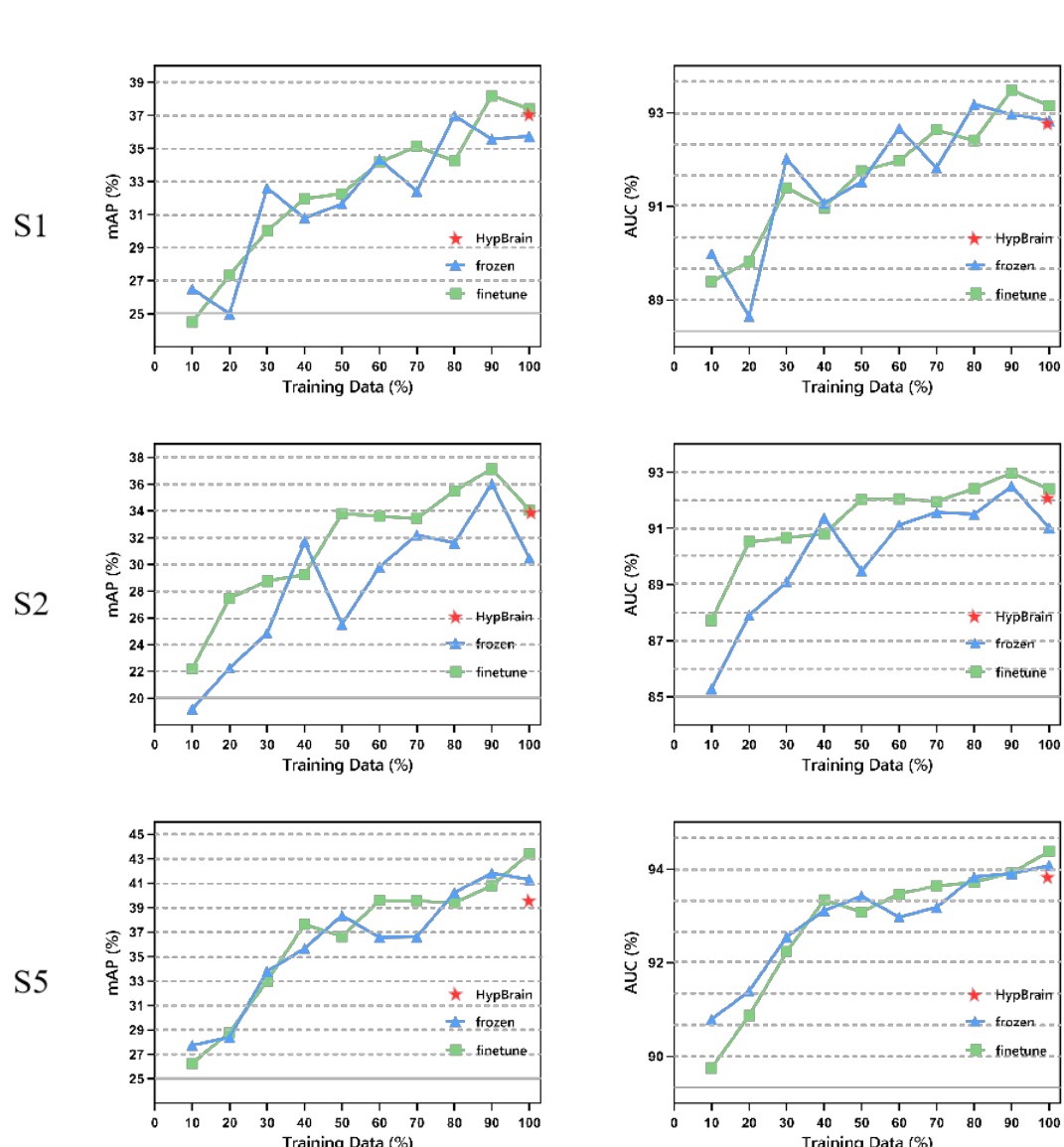

Figure 12: Cross-subject generalization of HypBrain-DeepSeek on unseen subjects in the multi-label prediction task.

