# OpenReview forum: "HypBrain: Hyperbolic Space Guided Cross-Subject Vision-Brain Representation Learning Framework"
_ICLR.cc/2026/Conference — ICLR 2026 Conference Withdrawn Submission_

### Official Review · Reviewer_3ddF · 2025-10-21

**Soundness:** 3
**Presentation:** 3
**Contribution:** 2
**Rating:** 2
**Confidence:** 4

**Summary:**

The paper presents Hyperbolic embedding space for fMRI decoding, i.e. retrieval and multi-class label prediction(detection).
The main claim is that this representation is beneficial for this task as it can capture well the commonalities and differences in subject brain responses.

**Strengths:**

- The paper's application of hyperbolic embedding to fMRI decoding is a novel and interesting contribution.
- The proposed approach demonstrates improved performance on two tasks compared to the baselines.

**Weaknesses:**

The main justification given for the approach is its empirical performance on the two tasks. However, the reported performance is significantly worse than current methods, such as UMBRAE (Xia et al., 2024) and Wills Aligner (Bao et al., 2025), which makes it uncertain if the proposed approach can outperform current leading methods.


Papers:
- Bao, G., Zhang, Q., Gong, Z., Zhou, J., Fan, W., Yi, K., Naseem, U., Hu, L., & Miao, D. (2025). Wills Aligner: Multi-Subject Collaborative Brain Visual Decoding. Proceedings of the AAAI Conference on Artificial Intelligence, 39(13), 14194-14202.

- Xia, W., de Charette, R., Öztireli, A. C., & Xue, J. (2024). UMBRAE: Unified Multimodal Brain Decoding. European Conference on Computer Vision (ECCV).

**Questions:**

- Do you think your approach can give performance gap when applied on state-of-the-art approaches?

---

### Official Review · Reviewer_FWnS · 2025-11-01

**Soundness:** 2
**Presentation:** 2
**Contribution:** 2
**Rating:** 4
**Confidence:** 4

**Summary:**

This paper focuses on fMRI visual decoding research, mainly addressing fMRI-to-visual-concept classification and fMRI-to-image retrieval tasks. The authors introduce hyperbolic space, including Lorentz neural networks and representation learning methods in Lorentz space. The model achieves good results on the classification task, but its performance on the retrieval task is unsatisfactory.

**Strengths:**

+ The manuscript is reasonably well-structured, and the formatting is clear.

**Weaknesses:**

1. In my view, this paper simply applies established Lorentz-space-based model designs and representation learning methods to the tasks of fMRI classification and fMRI–image retrieval. Although the authors emphasize the importance of hyperbolic space in the introduction, I still feel that the paper’s contribution and insight to the field are rather limited.

2. At present, fMRI visual decoding studies based on the NSD dataset almost inevitably involve the fMRI-to-image reconstruction task. However, this manuscript does not include any evaluation on that task, which I consider to be a limitation.

3. The baseline models compared in the manuscript (ICLR 2024, MM 2024) cannot be considered advanced, and in the retrieval task, the model’s performance is even far below that of the baselines. This raises concerns about the necessity and effectiveness of introducing hyperbolic space.

4. Some of the formulas in the paper may contain errors or are not clearly explained; see Question 1–3 below for reference.

**Questions:**

1. Why is 𝐹_{𝑖𝑛}^𝐸 in line 167 of dimension 𝑣+1? Why is there one extra voxel?

2. Are the superscripts 𝐸 and 𝐿 on 𝐹 meant to denote Euclidean space and Lorentz space, respectively? I can guess it, but I think the author should explicitly clarify their meanings.

3. The expressions for 𝐹_{𝑖𝑛}^L and 𝐹_{𝑚𝑖𝑑}^L in line 170 are strange — why do they simultaneously belong to both Lorentz and Euclidean spaces?

4. When the model receives an fMRI scan from a subject, how does it determine which Lorentz Tokenizer should be used for that fMRI?

5. Based on my understanding, does the baseline model EucBrain-CLIP use the pretrained CLIP semantic space? If so, what is the difference between EucBrain-CLIP and MindEye (Backbone + Projector)?

6. Could the authors consider introducing some more recent baselines for comparison? In the classification task, the strongest baseline compared — CLIP-MUSED — is from ICLR 2024, and in the retrieval task, the strongest baseline — Lite-Mind — is from MM 2024. I believe that in the past one or two years, there should have been additional methods worth including for comparison.

---

### Official Review · Reviewer_niWa · 2025-11-01

**Soundness:** 3
**Presentation:** 3
**Contribution:** 3
**Rating:** 6
**Confidence:** 3

**Summary:**

This article focuses on understanding the intricate mappings between visual stimuli and their corresponding neural responses. Specifically, it tackles vision-brain representation learning to align paired images and fMRI responses, within hyperbolic geometry so as to capture the hierarchical associations. The article introduces a novel vision-brain representation learning framework with carefully designed network and loss functions. The method delivers promising performance.

**Strengths:**

The motivation of the approach is clear. The proposed vision-brain representation learning framework is well designed, and novel to me. The core techniques are well justified in ablative experiments. The entire framework shows promising performance on public benchmarks.

**Weaknesses:**

- The rational behind introducing a dedicated hyperbolic tokenizer is unclear to me. Can we achieve tokenization more directly by applying a Lorentz Linear transformation, as used in the image branch? Moreover, there lacks an ablation study to verify the superority of the proposed tokenizer against alternative designs.

- Table 2 reveals substantial performance differences across various VLMs (CLIP, BLIP, DeepSeek). The question is what factors account for these disparities, why do certain VLMs excel while others underperform on this task? Additionally, it would be valuable to explore how performance scales with larger VLMs, or even closed-source models like GPT4o.

- How is the value of coefficient \lambda determined? How sensitive is the overall performance to the value?

**Questions:**

See Weaknesses

---

### Official Review · Reviewer_HG9t · 2025-11-03

**Soundness:** 2
**Presentation:** 2
**Contribution:** 3
**Rating:** 4
**Confidence:** 4

**Summary:**

The paper introduces HypBrain, a framework for cross-subject vision-brain representation learning that maps fMRI signals and image embeddings into a shared Lorentz manifold (hyperbolic space). The method uses a novel hyperbolic fMRI encoder and a combined hyperbolic contrastive and entailment loss.

**Strengths:**

The results are good. HypBrain consistently outperforms its direct Euclidean counterpart (EucBrain) in multi-label prediction and retrieval tasks. This strongly supports the central hypothesis that hyperbolic space is advantageous for semantic alignment in this domain.

The ability to achieve near full-data performance on a new subject (S7) with only 50% of the subject's data is a significant practical advantage for fMRI research, which is limited by data scarcity.

Figure 3 visually confirms the desired geometric separation: abstract Image embeddings cluster near the hyperbolic origin, while specific fMRI responses are pushed to the periphery.

Competitive Efficiency: HypBrain is highly competitive with powerful SOTA generative models (like MindEye), achieving high performance with a fraction of the parameter count (around 40M vs. 1000M+). This is well-argued as a focus on high-level semantic alignment over low-level pixel reproduction.

**Weaknesses:**

I have problems with section 2.1.1 and the equations.
Is Fin defined in line 170 as the projection of Fin in line 168 to hyperbolic space? Shouldn’t the dimension be the same one + 1 time dimension?
What is cin in equation 1?
Equation 1 follows Equation 16. However, why is a transform of Fin calculated first? I also don’t get where the projection to the Lorentz model via the exp map is performed. Then, in equation 2, to have the representation, there is a concat with a rescale of cmin and cin. But does the paper assume that Fmid is lying in a space with cmid curvature? Having the affine transform can break this assumption. Can the authors provide a clear mathematical derivation for these equations to show what space they are in and make sure that the final representation is a valid Lorentz representation in a space with cmid?

I also have a problem with equation 5. Comparing it with equation 21, why is cmid used? Can the authors provide elaborations on this equation? In line 210, what is Mmid? It is not defined.
Lorentzian MLP and the process to get Umid are unclear.

The paper motivates the use of hyperbolic space to capture hierarchical and is-a relationships. However, it does not include any analysis of the hierarchical structure within the learned space. Figure 3 indicates that the image representations are positioned closer to the origin, but no further examination of the space’s hierarchical properties is provided.


Would you please provide detailed, explicit information on the differences in the pipelines for EUCBrain and HypBrain?

I would highly recommend citing more hyperbolic-related work, such as Hyperbolic Deep Learning in Computer Vision: A Survey.


I have a suggestion for the title: HYPBRAIN: A Hyperbolic Space–Guided Framework for Cross-Subject Vision–Brain Representation Learning


It would be more beneficial to provide citations inside Table 1.
The sentence in line 337 does not have a meaning.
Line 54, why is the citation black?
Figure 1 : cat and cat and dog have the same distance from the origin?

**Questions:**

Please check the weaknesses.

---

### Note · Authors · 2025-11-27

I have read and agree with the venue's withdrawal policy on behalf of myself and my co-authors.